# Spatial Characteristics of Population Activities in Suburban Villages Based on Cellphone Signaling Analysis

**Jizhe Zhou** **, Quanhua Hou * and Wentao Dong**

School of Architecture, Chang'an University, Xi'an 710061, China; 2017041003@chd.edu.cn (J.Z.); dwt321654987@gmail.com (W.D.)

*   Correspondence: houquanhua@chd.edu.cn

**Abstract:** There are frequent population flow and complex spatial structures in suburban villages. Understanding the spatial characteristics of population activities in suburban villages helps to coordinate the relationship between urban and rural areas and guide the development of suburban villages and the formulation of sound policies. Taking the rural area of Qin and Han New City as the research object, this paper constructs a population time-space analysis framework of "population attribute-activity characteristics-spatial analysis" based on cellphone signaling data. According to the characteristics of the population activity curve, K-means clustering algorithm was used to classify rural space and analyze their characteristics. This study has shown that migrants, who are showed as young and energetic, account for 49.8% of the local registered population per day. Bidirectional flow of residents and commuters is generally presented in urban and rural areas. The urban-rural relation curve was characterized by "double peaks". The changes in the population in each village and the intensity of urban-rural relation were affected by location, industry and land use. The village population activity curve was classified into three categories, and nine characteristic villages are formed combined with the activity function. The research results can provide a scientific basis for urban and rural planning, spatial planning, industrial guidance and the facility layout.

**Keywords:** suburban villages; cellphone signaling data; population activity; spatial characteristics

## 1. Introduction

Suburban villages are close to cities. Compared with the other villages, the urban and rural transportation is more convenient, and material and energy exchange between various elements and their functions is quite frequent as a result of urbanization [1]. With the economical outward development of a multi-center urban development model, more suburban rural areas begin to be affected by urban expansion [2,3]. In order to accelerate the urbanization, a number of industrial development zones and new urban areas have been formed quickly in the suburbs of the city. The lands in the suburbs have been invaded, the social ecological environment has been seriously damaged, and the rural areas are faced with the dilemma of "passive urbanization" [4–6]. In order to further promote the development of regional economy, new national-level districts with larger scales have emerged in China [7]. Oriented by the policy, the suburbs are urbanized unnaturally, attracting a large influx of population, industry and capital [8]. Due to the leaping urbanization, the contradiction between urban and rural areas in suburban areas has been aggravated [9–12], which can also be reflected in the forecast of the population in rural areas, the diversity of migration purposes and the complexity of population flow. There are great differences between the actual population and the registered population in suburban rural areas, with differences in the amount and distribution in different periods of time [4].

The villages and their surrounding areas contain the floating population for diverse purposes, such as residence, employment, pension and entertainment. The connections with cities and their surrounding areas are complex [13–17]. The characteristics of suburban space are also more sophisticated because of population activities [18,19]. In order to better coordinate urban-rural relation, guide the development of suburban villages and promote the formulation of rational policies, the spatial characteristics of population activities in suburban villages need to be accurately grasped [20]. Qin and Han New City is one of the new cities in Xi Xian New District, with the urban built-up areas of Xi'an and Xianyang in the south. At present, rural areas account for 90%, which have the characteristics of frequent rural population movements, complex land use and diverse industrial forms. Compared with traditional surveys, such as questionnaires, interviews and tracking methods, this paper uses cellphone signaling data which has a larger sample with dynamic, precise and spatial attributes in order to study the spatial characteristics of local rural population activities [21,22]. Moreover, a suitable spatial and temporal analysis method is proposed.

## 2. Literature Review

With the advantages of large sample size, full coverage and continuous spatial and temporal attributes, cellphone signaling data can be used to study population activities and spatial characteristics. At the regional level, it is applied to regional urban systems [23], regional relations [24], tourist behaviors and preferences [25,26], etc. At the urban level, it is reflected in urban boundary demarcation [27], spatial structure [28,29], built environment evaluation [30], occupational and residential balance [31,32], population distribution [33,34], consumer behaviors [35] and spatial dynamic characteristics [34,36,37] analysis. At present, the existing research mainly focuses on the intercity, city and urban areas, while the research on rural areas is scarce.

In terms of the research methods of population activity spatial characteristics based on cellphone signaling data, there is a method of "activity intensity change + population composition change" according to a region division (Zhong and Wang, 2016) [34]. However, raster data are generated on the basis of the core density in actual research, which is only applicable to cities with high base station density, and there will be large errors in studies of rural areas. Since the temporal and spatial behaviors of suburban villagers are more complicated, the method needs to be further optimized. In addition, a "dynamic distribution - spatial activity" research framework in Shanghai's population distribution and spatial research is established by introducing the "day-night ratio" indicator [36]. The framework and indicator can also be applied to rural areas, but the identification of occupation and residence is not applicable to villagers' complex job-residence habits. Wang Yuzhuo established a quantitative model and function of population activity density and urban space to explore the internal mechanism of "population-space" (Wang, 2017) [38]. However, the indicators of population activity density can hardly apply to suburban villages with huge spatial changes in population. Therefore, this paper takes the group of villages in Qin and Han New City as an example and intends to use the cellphone signaling data to analyze the spatial characteristics of suburban rural population activities from the perspective of rural areas and population behaviors. The study is expected to provide guidance for synchronous development and planning in urban and rural areas.

## 3. Materials and Methods

### 3.1. Data Source

Figure 1 shows the region of Qin and Han New City. This paper mainly uses 2G, 3G and 4G cellphone signaling data of Qin and Han New City users in four weeks of 2018. The records generated by the cellphone actively and passively interacting with the base stations are its sources, including switching on and off, making calls, sending and receiving text messages, surfing the Internet, periodic location updates, location area switching, 3G conversion, 4G behavior, etc. These users included not only residents in suburban villages, but also visitors and commuters in Xi'an and Xianyang.

There was no information about individual attributes in the single piece of anonymous data. However, the big-sample data included population information such as the sex, the age group, the phone bill and the location. There were 88,000 identification numbers of different mobile phones in suburban rural areas recorded in this paper, accounting for 40% of the total resident population (statistical yearbook). The total number of daily records was about 35–40 million.

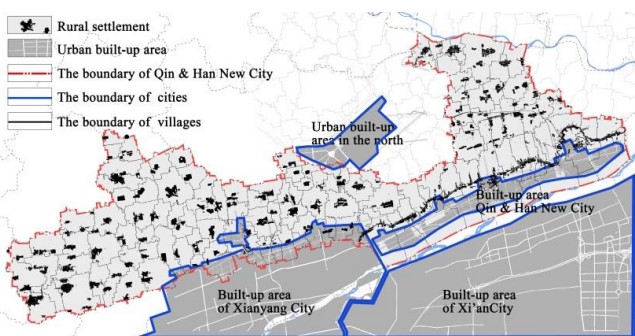

**Figure 1.** Urban-rural relations in Qin and Han New City.

The other data in this paper included the household registration data (including gender) of 142 villages in Qin and Han New City, which was collected from the census register departments of various towns and streets. The AutoCAD topographic map of the entire area (1:1000), which was derived from the Management Committee of Qin and Han New City. The construction situation, total area, cultivated land area, per capita income, road hardening rate, public service facilities (including primary schools, kindergartens, culture stations, clinics), historical and cultural resources, natural resources, distinctive industries and major occupational data in 142 administrative villages were derived from the compilation of basic materials in the village layout planning of Qin and Han New City in 2018.

*3.2. Analysis Framework*

Figure 2 shows the overall framework of "population attributes-activity characteristics-spatial analysis" established in the study. First of all, the population attributes were analyzed to determine the population staying purpose, population distribution and the age composition within the region. Secondly, the periodic characteristics of rural population activities were analyzed. To determine the characteristics of rural population activities, five types of indicators and three characteristics of overall activities were used. Finally, a clustering analysis was carried out on the spaces with the same activity characteristics, and a new understanding of the rural space was formed based on the space-time activities of villagers.

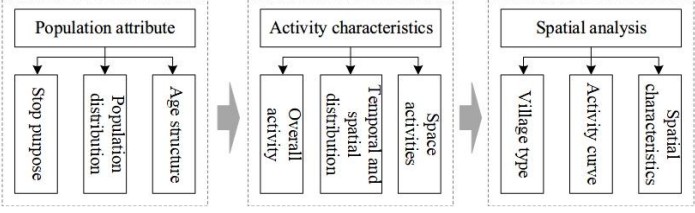

**Figure 2.** Analysis framework.

*3.3. Data Processing and Analysis Methods*

3.3.1. Positioning Method in Rural Area

In the study of urban areas, COO (Cell of Origin) positioning technology was the most commonly used [39]. However, the distance between each base station in the suburbs was 500–2000 m, and the

COO positioning technology could cause huge errors. Therefore, this study divided the entire domain by 4871 grids of 250 m × 250 m, and the base station handover positioning technology was added on the basis of COO. The weight of the handover numbers of the base stations and the surrounding base stations was used to jointly locate the grid where the user was located [40]. Specifically, based on the monthly cellphone signaling data of users, their behaviors could be divided into stay and travel, and the changes between the stay points were identified as travel. Therefore, the location of the resident point was particularly important. The study on determining the stay point was accomplished joint with the "Intelligent Footprint" company and divided into four steps showed in Figure 3. Firstly, we divided the research area into some square "base areas" (which may overlap) according to the location, power and coverage of the surrounding base stations. There were about three to five base stations in or around the "base area" generally. Secondly, we identified the resident behavior of users. Specifically, when a user accessed a new base station and continued to contact the next, which was within the "base area" or frequently cutting back to the first base station (signal overlap). If this state remained unchanged for half an hour, the user was identified as a "stay point" initially. The start time was the time when they first accessed the base station, and the location was identified within the "base area" initially. Thirdly, the more accurate coordinates of the stay points were obtained by interpolating the location centroid algorithm according to the frequency of communication between the users and the distance of their surrounding base stations [41]. Finally, the location of the stay points was corrected through a month's observation gradually. Theoretically, the location, which was generally located at a fixed grid, changed little over a long period of time [41]. In order to improve the positioning accuracy, the region was divided into different identification units, so that the users' stay and travel behaviors were identified through the dynamic observation of the switching events and time between different base stations in the cells [42]. Finally, the judgment of the previous month was revised through observation and judgment in a later month to optimize the observation results. Since rural settlements clustered inside and scattered outside, residents were mainly gathered at the settlements at 0:00. In order to verify the positioning accuracy, this study defined grids with the number of people greater than 100 as a hotspot grid. By comparing and checking regional hotspot grids at 0:00 with the actual settlement locations of 142 villages, it was found that the average distance between the settlement locations and the grid hotspots was about 400 m. Among them, 73.6% of the positioning errors were less than 500 m. There were three types of rural settlements with large positioning errors: The first type was the rural settlements located in the urban fringe area. The second type was the rural settlement with industrial and commercial services. The third type was the settlement located in the area of sparse base stations. However, the spacing of rural settlements in the research area was generally greater than 1000 m. Therefore, the positioning accuracy was acceptable.

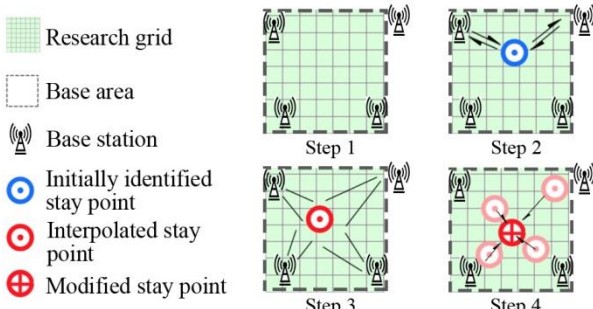

**Figure 3.** Positioning method of cellphone signaling.

### 3.3.2. Classification and Identification of Population Attributes

In the analysis of rural population attributes, the age composition was divided into seven intervals and their proportions were calculated respectively. The research objectives were divided into three categories: "Resident population", "temporary resident population" and "external commuting

population". In order to verify the rationality of the data, it was necessary to test the linear correlation among age data, resident population data and rural field survey data. The correlation coefficient was 0.52, which showed a strong correlation. This accuracy can was acceptable considering inconsistency in the proportion of sampling statistics and Unicom users in various villages. However, the data about sex and consumption were not used because of the low linear correlation coefficient with the actual survey data.

Since there were a large number of permanent residents who were not registered in the local area, it was necessary to identify the "resident population" in villages so as to analyze the purpose of staying. Whether the user was a local resident should be judged according to the standard of staying more than 10 days within 30 days in the local area [43]. The permanent residence was judged, namely, the seconds that the user was observed from 21:00 in a place to 8:00 the next day, was accumulated and ranked monthly. The highest ranked area was the user's permanent residence, and the user was identified as "resident population". In this paper, the "temporary resident population" was identified by subtracting the number of resident population from the average number of people observed from 0:00 to 6:00 in the morning. After excluding these two types, the population of the remaining time was identified as "external commuting population".

### 3.3.3. Data Processing and Indicator Construction

(1) Overall activity curve of villages

The records of the total population and resident population in 20 working days and eight weekends were counted in units of 1 hour, and the total number of records in each period was calculated. The formula is as follows:

$$T_j = (O, P, R, Q)_j = \sum_{k=1, i=1}^{k,i} T_{i\_j} \tag{1}$$

where $k$ is the grid number ($k = 1, 2, \ldots, 4871$), $T_{i\_j}$ is the number of population records of $k$ grid in period $j$ of the day $i$($i = 1, 2, \ldots, 28$, $j = 1, 2, \ldots, 23$), $i$ is the date ($i = wi, wdi$) where $wi$ represents the weekday number ($wi = 1, 2, \ldots, 20$) and $wdi$ represents the weekend number ($wdi = 1, 2, \ldots, 8$), $(O, P, R, Q)_j$ represents the population records in period $j$ (the total population of the weekday, the total population of the weekend, the resident population of weekday and the resident population of weekend). The curves of different population activities at a different time in the rural areas are expressed as:

$$\{(O, P, R, Q)_0, (O, P, R, Q)_1, \ldots, (O, P, R, Q)_j, \ldots, (O, P, R, Q)_{23}\} \tag{2}$$

(2) Day-night ratio indicator

According to the local habits, the ratio of the daily average total population records at 0–5 o'clock to the daily average records at 10–15 o'clock was used as the day-night ratio indicator based on each grid. The formula is as follows:

$$Index_{k\_day\_night} = (U, V)_{k\_day\_night} = \frac{T_{day\_6}}{T_{night\_6}} \tag{3}$$

where $T_{day\_6}$ and $T_{night\_6}$ are the total population records of the grid at 10–15 o'clock and 0–5 o'clock, respectively, and $(U, V)_{k\_day\_night}$ is the day-night record ratio of grid $k$ in (weekdays, weekends).

(3) Activity intensity of villages

An "activity" defined in this study was that of a user starting from origin O and the location that they reached or farthest reached was the destination D within half an hour. The number of activities identified as "activities" in each village within four weeks was summarized and counted, with a total

of 672 h. If 1 hour as a time unit, an activity intensity statistical vector $T$ is constructed. The formula is as follows:

$$T_{l\_672} = (O,D)_{l\_672} = \{T_{w1\_0}, T_{w1\_1}, \dots, T_{wi\_j}, \dots, T_{w20\_23}, T_{wd1\_0}, T_{wd1\_2}, \dots, T_{wdi\_j}, \dots, T_{wd8\_23}\} \quad (4)$$

where $l$ is the village number ($l = 1, 2, \dots, 142$), $T_{wi\_j}$ is the number of records of the village during the period $j$ in the weekday $wi$ ($wi = 1, 2, \dots, 20$), $j$ is the start hour ($j = 1, 2, \dots, 23$), $wdi$ is the serial number of the weekends ($wdi = 1, 2, \dots, 8$) and $(O,D)_{l\_672}$ indicates the quantity of recording "activity" with a village as (origin, destination). Through the correlation analysis of 28-day activity record data, the results showed that whether the origin or the destination showed a significant correlation at the 0.01 level in the same time period on all workdays and at weekends. Therefore, the amount of "activity" of all weekdays and weekends could be averaged to obtain the average number of records in 24 h per day on weekdays and weekends, and the data dimension was simplified. The formula is as follows:

$$X_{l\_48} = \{X_{w\_0}, X_{w\_1}, \dots, X_{w\_j}, \dots, X_{w\_23}, X_{wd\_0}, X_{wd\_1}, \dots, X_{wd\_j}, \dots, X_{w\_23}\} \quad (5)$$

where $X_{w\_j}$ is the average number of records in period $j$ on weekdays and $X_{wd\_j}$ is the average number of records in period $j$ on weekends.

(4) Indicator of urban-rural connecting

The ratio of the total contact quantity of each village and urban area (Xi'an, Xianyang and northern urban areas) in 28 days was summarized. The indicator is $R\_avg_l$, and the formula is as follows:

$$R\_avg_l = (C,V)\_avg_l = \frac{\sum_{i=1}^{i} X_{od\_i}}{\sum_{l=1,i=1}^{l,i} X_{od\_i}} \quad (6)$$

where $X_{od\_i}$ is the quantity of the urban-rural connecting of the village $l$ ($l = 1, 2, \dots, 142$) on day $i$ and $(C,V)\_avg_l$ indicates the proportion of village connecting quantity in the total connecting quantity starting from (city, village). The ratio of the total rural population to urban connecting quantity at each period of the day was defined as $R\_avg_l$, and the formula is as follows:

$$R\_avg_j = (C,V)\_avg_j = \frac{\sum_{l=1,i=1}^{l,i} X_{odi\_j}}{\sum_{l=1,i=1,j=1}^{l,i,j} X_{odi\_j}} \quad (7)$$

where $X_{odi\_j}$ is the number of urban-rural connecting records in period $j$ of day $i$ ($i = w, wd$) and $(C,V)\_avg_j$ indicates the proportion of urban and rural connecting quantity in each period to the population of that in a day when the origin of the connecting is (village, city). The urban-rural connecting curve is as follows:

$$\left\{ (C,V)_{avg_0}, (C,V)\_avg_1, \dots, (C,V)_{avg_j}, \dots, (C,V)\_avg_{23} \right\} \quad (8)$$

(5) Intensity ratio of weekdays and weekends connecting

For each village, the calculation formula for the ratio of the daily average OD records of the 4-day weekends to the average OD records of 20-day weekdays is as follows:

$$Index_{l\_wd\_w} = \frac{\sum_{j=1}^{j} X_{wd\_j}}{\sum_{j=1}^{j} X_{w\_j}} \quad (9)$$

where, $l$, $j$, $X_{w\_j}$ and $X_{wd\_j}$ have the same meanings as those in Formulas (4) and (5).

### 3.3.4. Spatial Clustering Method

Combined with the status quo of rural land use, each village was classified based on the periodic characteristics of rural spatial activities over time. The characteristics of rural space could be summarized. At present, spatial clustering methods using mobile phone signaling included proximity similarity propagation clustering algorithm [44], K-means clustering algorithm [45], fuzzy C-means clustering algorithm [46], DBSCAN density clustering algorithm [47], etc. The fuzzy C-means clustering algorithm and the proximity similarity propagation clustering algorithm are not as accurate as the K-means clustering algorithm when dealing with high-dimensional complex data [48]. The DBSCAN density clustering algorithm with higher precision is generally used for density-based region partitioning, and it is more suitable for raster data or kernel-density data [49,50]. In this paper, the K-means clustering algorithm was adopted, which has the characteristics of obvious clustering effect and high computational efficiency. It is more suitable for the data with the number of trips and Gaussian distribution as the original data set. At present, the K-means clustering algorithm has been widely applied in the research of express delivery, travel characteristics and clustering stay points based on the OD point and the OD flow data sets [51–53]. The K-means clustering algorithm was used to calculate the appropriate K value. The number of clusters was estimated by using the elbow rule [54]. In addition, the rural activity intensity indicator data and the urban-rural flow indicator data were selected as the original dataset of the cluster.

## 4. Analysis Results

### *4.1. Suburban Rural Population Attributes*

#### 4.1.1. Staying Purpose

The number of people staying in rural areas is shown in Table 1. The number of permanent residents in suburban villages was 1.2 times that of the registered population, including not only the rural settlement population, but also the residential areas, industrial parks and resettlements population. In addition, there were a large number of temporary residents and commuters. When these two types of population were regarded as a floating population, the peak appeared at 14:00 on the weekdays, accounting for 49.8% of the registered population.

**Table 1.** Staying purposes of rural population in Qin and Han New City.

| Purpose | Registered Resident | Permanent Resident | Temporary Resident | The Largest Commuting Population Per Day |
|---|---|---|---|---|
| Population (thousand) | 221 | 265 | 51 | 59 |

#### 4.1.2. Regional Population Distribution

Figure 4 shows the distribution of permanent residents, and there was 42% of whom were distributed in the vicinity of the urban built-up area of Xianyang City and on both sides of the National Highway 312 extending in the northwest direction. The floating population was also gathered in these areas, and the population on both sides of the road was denser on weekdays. This study found that the proportion of rural construction land around the New City built-up area was much larger than that in the Xianyang urban fringe area. However, there was not obvious population aggregation in the new urban area, and the urbanization rate was 35.1%. Rural urbanization takes the lead in the vicinity of Xianyang city and follows the traditional "point-axis" model, which was less affected by the new city.

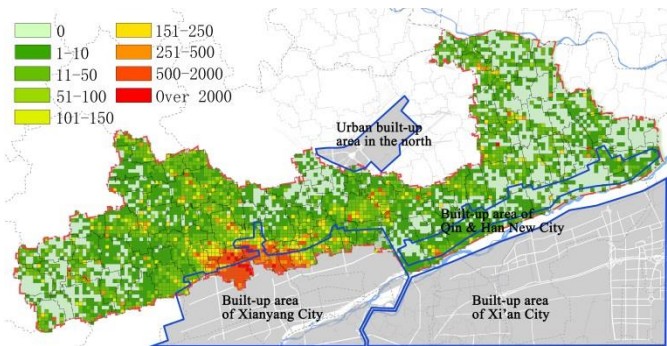

**Figure 4.** Distribution of permanent residents.

### 4.1.3. Population Age Composition

Compared with the age groups of Xi'an and Xianyang, it can be seen from the Table 2 that the age composition of the rural population was not much different from that of the surrounding cities. However, the proportion of the population aged 19–60 was much higher than that in other rural areas of Shaanxi Province.

**Table 2.** Age composition in Qin and Han New City and surrounding cities.

| City | Age Composition (%) | | | | | | |
|---|---|---|---|---|---|---|---|
| | <18 | 19–30 | 31–40 | 41–50 | 51–60 | 61–70 | >70 |
| Xi'an | 3.92% | 34.37% | 23.53% | 15.29% | 8.66% | 3.61% | 1.17% |
| Xianyang | 4.83% | 30.60% | 22.00% | 18.01% | 9.25% | 4.11% | 1.55% |
| New City | 4.84% | 32.54% | 21.72% | 18.62% | 9.59% | 4.22% | 4.52% |

In general, the suburban rural population structure was complex, and the resident population was 1.2 times as large as the registered population. The proportion of the floating population was large. The gathering of the rural population was mainly driven by the urbanization of Xianyang. The overall age of the rural population was younger.

### 4.2. Characteristics of Rural Resident Activities

### 4.2.1. Overall Characteristics

The overall activities of the suburban rural areas are shown in Figure 5. On weekdays, 0:00–6:00 was usually sleeping time and the total rural population changed little. External commuting population began to enter the area from 6:00 and reached a relatively stable state at 11:00. The largest commuting population was about 59,000. The population gradually decreased after work at 17:00 and the commuting population gradually left this area after 22:00. Since the resident population returned and eventually stabilized at 0:00, a small increase occurred. The residents commuted from villages to cities at 7:00 and reached a relatively stable state at 12:00. The resident population working in the city was about 18,000, and a small part of them returned after work at 17:00. After 18:00, some rural people entered the city for work and entertainment, and most of them returned after 22:00. In addition, about 4000 permanent residents were night workers, who returned from 22:00 to 7:00 the next day. The flow features of the population at weekends were similar to the weekdays, with a maximum commuting population of about 49,000. There were 3% of suburban residents who lived in the city at weekends, and they had the characteristics of "urban and rural amphibious". These characteristics were generally consistent with the behavior of the villagers in the field survey, which could also be reflected in the bidirectional flow of residents and commuters in urban and rural areas.

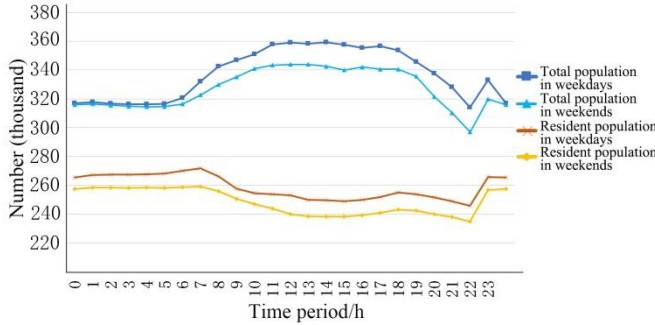

**Figure 5.** Overall activity curve in Qin and Han New City.

### 4.2.2. Temporal and Spatial Distribution Characteristics

Figure 6 shows the main employment, tourism and residential areas in Qin and Han New City. The day-night ratio indicators are shown in Figures 7 and 8. Since the average day-night ratio of each grid was 1.83, it was considered that 1.6 to 2.0 was the equilibrium state. The ratio was more balanced in suburbs on weekdays than weekends. The employment areas have a high day-night ratio, such as suburban industrial parks, agricultural parks, large amusement parks and the thermal-power plant. The areas with relatively low day-night ratio were usually densely populated settlements. On weekends, the day-night ratio increased significantly in areas with leisure service functions, such as Qin Dynasty Museum, the Han Yang Mausoleum and Chateau Castle. In addition, the ratio was lower in areas where the residential function was dominant, which also reflected the emigration of local villagers.

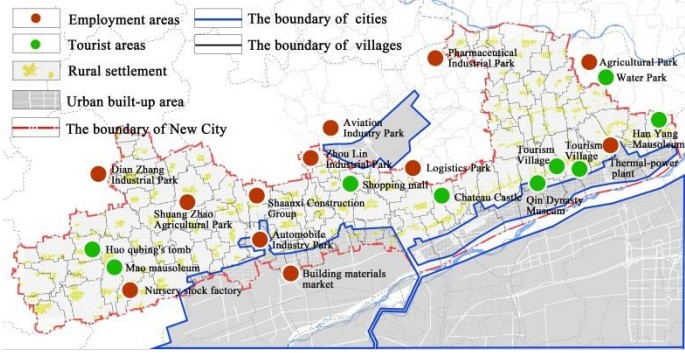

**Figure 6.** Location of main functional regions.

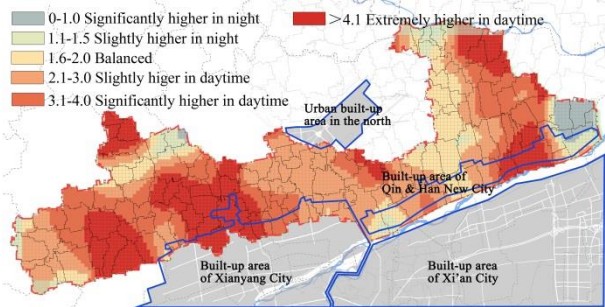

**Figure 7.** Day-night ratio distribution on weekdays.

Figure 9 shows the spatial distribution of rural activity which had three characteristics: Firstly, the intensity of rural activities gradually increased from 5:00, reaching two peaks at 8:00 and 18:00, respectively; secondly, rural activities increased obviously in the urban fringe areas; finally, the intensity

was influenced by the industrial and commercial areas. In addition, it was found that the intensity of population activity in suburban villages was not remarkably related to population density, and the areas surrounding New City with low population density also had strong vitality.

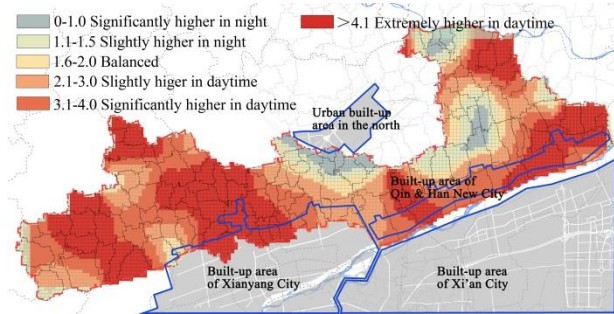

**Figure 8.** Day-night ratio distribution on weekends.

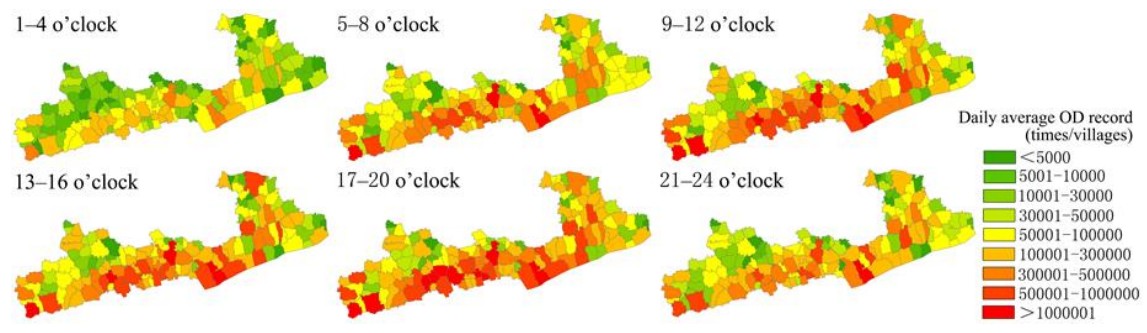

**Figure 9.** Activity intensity within six hours of each village on weekdays.

### 4.2.3. Characteristics of Spatial Activities

From the urban-rural connecting indicator, it could be found that the records of urban visiting villages and rural visiting cities differed by only 1.1%. The connections between villages and Xi'an, Xianyang and northern urban areas accounted for 53.2%, 29.3% and 17.4% of the total, respectively. Although villages were all near small cities, bigger cities had more connections.

Figure 10 shows the connection features of rural visiting cities. The two peak periods were 8:00 and 18:00. The morning peak of weekdays was relatively earlier and its duration was shorter. The working time from 10:00 to 16:00 was relatively stable; after 16:00, the second peak appeared, and the connections of peak-time accounted for 10% of the total. Peaks appeared relatively late at weekends. The connection feature is shown in Figure 11, which also presents "double peaks", but the peak period appeared at 8:00. The main working place in local areas was in suburban rural areas, and it was more urgent for commuting on weekdays.

The connecting intensity between villages and cities is shown in Figure 12. Firstly, villages with large-scale businesses, scenic spots and industrial parks had high connection strength between urban and rural areas. Secondly, villages with convenient traffic connection to cities also had high connecting intensity. Figure 13 shows the intensity ratio of weekday and weekend connecting, the connection of 82.3% villages on weekends was less than that on weekdays. There was no industry or agriculture in villages with strong urban-rural correlation on weekends, but resources were abundant.

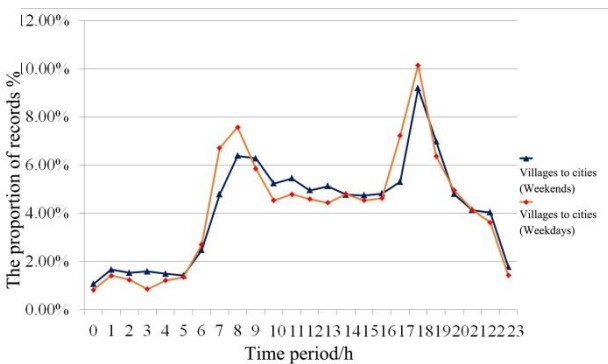

**Figure 10.** Connection curve of rural visiting cities.

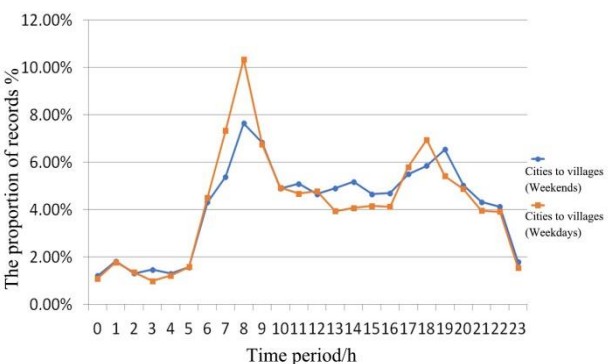

**Figure 11.** Connection curve of urban visiting villages.

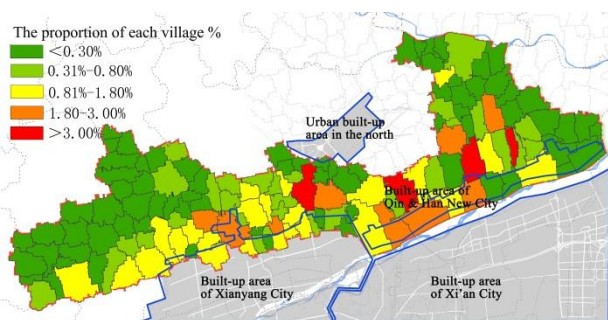

**Figure 12.** Connection intensity proportion of villages.

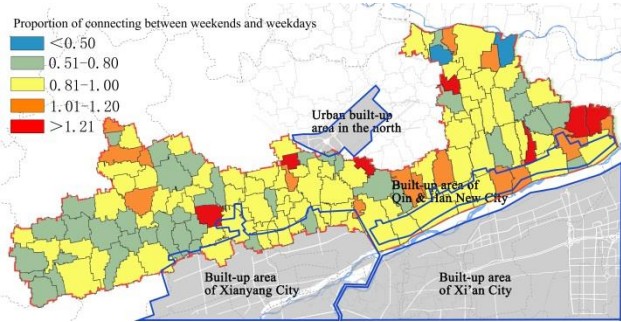

**Figure 13.** Connection intensity ratio between weekdays and weekends.

*4.3. Analysis of Suburban Rural Space*

Based on the analysis of the rural population activities, this paper attempted to classify suburban rural space from the perspective of the temporal and spatial distribution of the rural population and the characteristics of urban-rural linkage. In order to guide the planning and construction of local villages, administrative villages were defined as basic units. In the clustering process in Python software [55], the villages in Qin and Han New City were classified into nine categories of villages under three types of activity curves (the service-oriented villages were not subdivided due to their low external connection and vitality). The activity curves, main land use and industries corresponding to various villages were compared and analyzed (Figure 14).

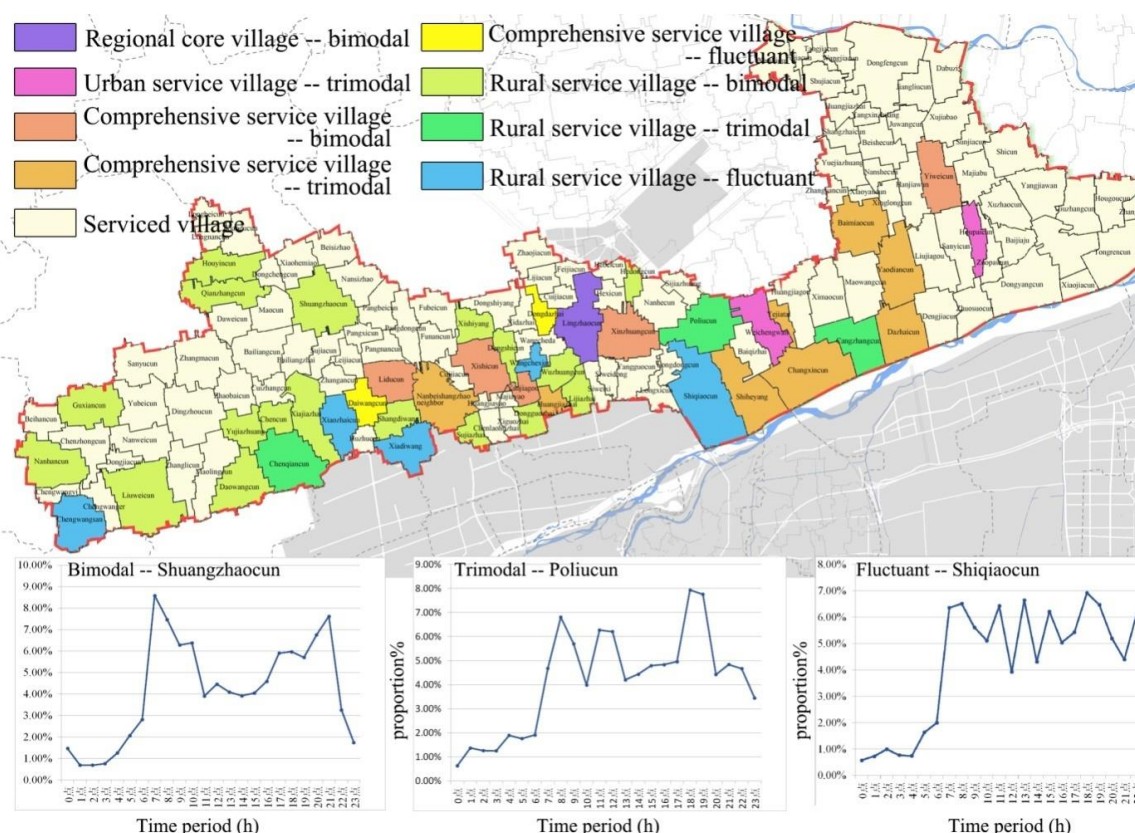

**Figure 14.** Spatial clustering results and typical activity curve waveforms.

(1) Core village. Core villages have much more connections with cities and villages than other types of villages. Such villages are usually located in suburbs, and the population activity curves are bimodal. There are two activity peaks in the morning and evening. The resident population of core villages is moderate, but the urban-rural connection is strong. The land in the village is mainly used for residential, commercial and public service facilities. There are regional-level commercial facilities and service facilities. That is, the dominant industries are commercial and service industries.

(2) Urban service village. Urban service villages have more connections with cities, but they are weakly connected with villages. They are usually located in the urban fringe area and the population activity curves are trimodal. Namely, there will be another activity peak between 11:00 and 14:00 except for morning and evening. However, the resident population is small. The construction land in the village accounts for a high proportion, mainly for residential and commercial use. There are a large number of tourist attractions in and around the villages, and service is the leading industry.

(3) Comprehensive service village. This type of village has more connections with urban and rural areas. The activity curve is classified into three categories. "The bimodal village" is generally

located in the suburbs of the city, and the resident population is moderate. Generally speaking, there is characteristic industry in comprehensive service villages. The proportion of construction land is higher, and the types of land use include industrial and residential land. "The trimodal village" is generally located in the urban fringe area, with a moderate resident population and a relatively high construction land in the village. Residential and commercial land is the common use type, and the leading industries include commercial and service industries. "The fluctuant village" is located in the suburbs of the city and has a large resident population. There is no obvious peak activity in a day, showing fluctuation. The leading industries are agriculture and agricultural sideline. There is moderate construction land, and the land is mainly used for residential and public service facilities. Some villages also have resettlement communities.

(4) Rural service village. This type of village has more connections with surrounding villages, and the connection with cities is weak. According to the activity curve, it is subdivided into three categories. With moderate resident population, "bimodal villages" are located in the regional center of suburbs or suburbs of cities. The proportion of land use is between the comprehensive service village and the serviced village, which is mainly residential. The leading industry is agriculture with a small amount of sideline and industry. "Trimodal villages" are located around the scenic spots on the edge of cities. The land is mainly used for residential and tourist facilities. Agriculture is the leading industry, and accommodation is provided for surrounding attractions. "Fluctuant villages" are located in the suburbs of cities and have large resident population. The area of construction land is small, and the land use type is mainly residential land. The leading industry is agriculture.

(5) Serviced village. This type of village generally has few external connections. Little land is used for construction, and agriculture is the leading industry. Some public services need to be provided by the outside areas.

In general, rural population activities are closely related to rural spatial characteristics. The existence of large-scale commercial and public service facilities in villages can greatly strengthen the external contacting ability. Villages dominated by primary industry and secondary industry generally have a bimodal activity curve. The third peak of population activity appeared between 11:00 and 14:00 in villages with the service function. In the villages with more permanent population, there was fluctuation in the population activity curve (Table 3).

After analyzing the activities and spatial characteristics of rural population, effective guidance can be provided for its spatial structure and industrial orientation. For example, core villages can be prioritized as areas for relieving urban population and function, developing commercial, service industries and driving regional development. Urban service villages should show relative heterogeneity in industrial development. Comprehensive service villages are generally the areas where urbanization takes the lead and undertakes the function of urban-rural connection. Rural service villages can be used as central villages which can provide good rural service functions.

**Table 3.** Analysis results of rural spatial characteristics in Qin and Han New City.

| Type | Connecting Intensity | Activity Curve | Distribution | Permanent Population | Proportion of Construction Land | Main Land Use Type | Leading Industry | Remark |
|---|---|---|---|---|---|---|---|---|
| Core village | Obviously strong with urban and rural areas | Bimodal | Suburban area | Moderate | Very high | Residential, commercial, public service | Commercial land service | Regional-level facilities |
| Urban service village | Strong with urban and weak with rural areas | Trimodal | The urban fringe area | Low | Very high | Residential, commercial | Service | Attractions around |
| Comprehensive service village | Strong with urban and rural areas | Bimodal | Suburban area | Moderate | Above average | Residential, industrial | Secondary industry | Characteristic industry |
| | | Trimodal | The urban fringe area | Moderate | Above average | Residential, commercial | Commercial land service | – |
| | | Fluctuant | Suburban area | High | Average | Residential, public service | Agriculture, sideline | Some with resettlement |
| Rural service village | Strong with rural areas and weak with urban | Bimodal | Centre of outer suburbs, Suburban area | Moderate | Average | Residential | Agriculture, with a small amount of sideline and industry | – |
| | | Trimodal | Suburban area | Moderate | Average | Residential, tourist service | Agriculture, with a small amount of accommodation | Attractions around |
| | | Fluctuant | Suburban area | High | Below average | Residential | Agriculture | – |
| Serviced village | Weak with urban and rural areas | – | Outer suburbs | Low | Very low | Residential | Agriculture | Lack of public services |

## 5. Conclusions and Discussion

### 5.1. Discussion

#### 5.1.1. Application of Cellphone Signaling Location in Suburban Villages

In suburban villages, population movements are frequent and complex, making it difficult to count the resident population. A method with larger samples and higher dynamic that should be proposed to solve the problems, and analyzing attributes and temporal-spatial behaviors accurately. It can be seen from the research that cellphone signaling has important value in the research field of rural population behavior dynamic analysis. Based on the cellphone signaling data, the attributes and spatio-temporal distribution information of rural population could be obtained more comprehensively, dynamically and accurately, which provides a better method for analyzing the dynamic characteristics of rural population. However, there are still problems of recognition accuracy in the application in rural areas. In the area with the highest recognition accuracy, there was a large distance between villages or a village near to only one base station. However, most rural settlements and base stations did not match in space, which would inevitably lead to identification errors. Although the COO technology combined with base station handover positioning technology and location centroid algorithm could reduce errors, it was only effective in the rural areas where the settlements with internal aggregation and external dispersion. If the distance between villages and settlements was less than 1000 m, the recognition error would increase significantly.

#### 5.1.2. Application in Town and Village Planning

From 2017 to 2018, our team completed the delegation of urban-village system planning in 76 of 142 villages in Qin and Han New City. This study played an important role in town and village planning. Firstly, through the study of the attributes of the rural population, the true quantity and distribution of rural residents and the floating population was successfully found, which provided the basis for planning. Secondly, by studying rural resident activities, the reasons for the abandonment of rural public service facilities were explored. The location of facilities was optimized, and the scale was predicted. The work has promoted the sharing of urban and rural public services. For example, 28 of the original 44 primary schools were optimized, and the investment in educational resources decreased substantially. Furthermore, 21 schools were established in the future urban regions to provide high-quality urban education for villagers. Thirdly, in the planning of village system, the central villages chose from "comprehensive service village" and "rural service village", and formed a "city-central village-village" three-level structure. A "urban-rural" two-level structure is formed in the areas with "urban service village". In addition, the "core village" would be an important choice for the deputy center of cities. The number of the original central villages was reduced from 14 to 8, and smart and equal allocation of urban and rural resources was achieved. Finally, based on the intensity of rural activities, urban-rural connections and spatial types guided the orientation, construction and industry of villages.

#### 5.1.3. Application in Planning Evaluation

At present, planning evaluation based on mobile signaling mainly focuses on three directions: Urban spatial vitality [30,38], land development potential [56] and public facilities service capacity [57]. This study provides an evaluation concept for rural planning. Space-time behaviors of the rural population were affected by the rural industry and land use. Namely, a reasonable industrial and spatial planning can provide benign guidance for the production and life in villagers. Studying the space-time behaviors of the rural population can check the rationality of the rural planning and optimize the spatial planning, industrial guidance and the facility layout. The results can not only question the rationality of the established rural planning, but also provide guidance for future planning and rational evaluation. In addition, the research optimizes the previous urbanization evaluation index

based on the connection density [38] by introducing the vitality intensity index. For the evaluation of the urbanization quality of Qin and Han New City as a national-level district, the result showed that it had not reached the ideal urbanization level. There was the possibility of forming a "ghost city", which confirms the conjecture of substandard urbanization of some new national-level districts by Dr. Xie [58].

### 5.1.4. Application of Rural Vitality Intensity

First of all, for the definition of spatial vitality based on cellphone signaling, mainstream research takes the population density as the leading indicator whether in the regional urban system or at the urban level [28,31,34,35]. However, there was no inevitable connection between population density and the vitality in suburban villages. Secondly, the vitality intensity of rural areas was more directly affected by regional cities. Villages with low population density but closer urban-rural correlation were more active than others. Therefore, this study used the amount of rural external connections as the criterion for the intensity of rural vitality. To some degree, the urbanization region of the suburbs in the future could be predicted by the research of the rural vitality intensity.

### *5.2. Conclusions*

Taking Qin and Han New City as the research object, this study innovatively analyzed the characteristics and activities of rural population as well as rural spatial classification combined with cellphone signaling data. The results showed that the number of permanent residents in suburban villages was 1.2 times that of the registered population. The peak of commuting population was 59 thousand and appeared at 14:00 on weekdays. The maximum floating population per day accounted for 49.8% of the registered population. About 42% permanent residents were distributed in the vicinity of the urban built-up area of Xianyang City and on both sides of National Highway 312. However, there was no obvious population aggregation in the new urban area, and the urbanization rate was only 35.1%. It was generally presented as a bidirectional flow of residents and commuters between urban and rural areas, and 3% of residents had the characteristics of "urban and rural amphibious". The urban-rural connection curve was characterized by "double peaks" from 8:00 and 18:00 with the connections accounting for 10% of the total. The connections between villages and Xi'an, Xianyang and northern urban areas accounted for 53.2%, 29.3% and 17.4%, respectively. The temporal-spatial variations of the population in each village and the intensity of urban-rural relation were affected by location, industry and land use. The connection of 82.3% villages at weekends was less than that on weekdays. Moreover, based on the urban-rural linkage features and the rural activity curve, the K-means clustering algorithm was used to divide the suburban area into nine characteristic villages under the three activity curves of five service categories. It was preliminarily deduced that suburban rural population activities were closely related to rural spatial characteristics.

Compared with the previous studies. First of all, intercity and urban studies can be combined by taking suburban villages as objects to form an urban and rural population time-space behavior analysis system. It is an extension and supplement of the previous research area. Secondly, the research aimed at the unnaturally formed "passive urbanization" villages, and solved the problems of population prediction, migrating purpose and the flow direction. Moreover, by analyzing the population behavior, the spatial characteristics of villages matched initially. The findings can provide reference and basis for relevant studies, such as time-space behavior of villagers, spatial structure optimization, urban-rural coordination, rural policy making, the urbanization prediction, effectiveness evaluation of national new district and the impact of suburban industrial parks and urban sub-centers. Finally, exploring the effectiveness evaluation, location method, clustering method and rural vitality judgment can provide a reference for subsequent studies. Our team will carry out further research on urbanization, the urban-rural relationship and urban-village system. However, there is a lack of mechanism research on the matching between rural spatial characteristics and time-space behavior of villages. However, there

are relatively few objects in this paper, leading to the difficulty of universal research and induction. Also, the description of rural spatial characteristics still needs more accurate and comprehensive data.

**Author Contributions:** Conceptualization, J.Z.; Formal analysis, J.Z.; Investigation, W.D.; Project administration, Q.H.; Supervision, Q.H.; Writing—original draft preparation, J.Z.; Writing—review and editing, Q.H. and W.D.

**Funding:** This research was funded by the Major Theoretical and Practical Research Project in the Social Sciences in Shaanxi (Grant No. 2018Z025), Ministry of Culture and Tourism Research Project (Grant No. 18TABG019) and the Fundamental Research Funds (Social Science) for the Central Universities of China (Grant No. 300102419631).

**Conflicts of Interest:** The authors declare no conflict of interest.

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
