# Peer review of "Spatial Characteristics of Population Activities in Suburban Villages Based on Cellphone Signaling Analysis"

_sustainability, doi:10.3390/su11072159_

Round 1
Reviewer 1 Report
I see some method merits of this paper but it is writing in a very un-academic way. The paper does not engage theoretical background and enough literature thus I cannot see where the contribution of their works lies. Typical papers in this field are usually with 400-500 lines length and with 40-50 references. This paper is very short and lacks citations not because it is concise—but because the authors fail to provide in-depth discussions. I think the authors are new-comers for academic writings and would like to suggest some revisions to help because there are some merits in the methods. First, the introduction, discussions, and conclusions need total re-writing to better engage in the academic background of your research. Also, I suggest the authors adding a literature review section to show you have read related contributions and make clear where you add to these studies. I would like to suggest the following paragraphs to be added:
First, discuss the phenomenon of urbanization, urban growth and the mechanisms of interaction between urban CBD, sub-centers, and suburbs, for example:
https://www.sciencedirect.com/science/article/pii/S0264275118312022
Second, discuss the previous international literature on suburban and rural migrations:
https://link.springer.com/article/10.1007/s00168-009-0325-4
Third, discuss how the migration is related to suburban growth and how mobile phone data can contribute to the studies:
https://www.sciencedirect.com/science/article/pii/S1364815203000306
Finally, please make a thorough review on how mobile phone data is used in your related research field, for example:
https://www.sciencedirect.com/science/article/pii/S0968090X09000400
Other detailed comments:
Line 9: “If the spatial characteristics of population activities cannot be accurately grasped, the research on rural space, land use and industry will be hardly launch.” This is a very strong and ungrounded argument. Please consider removal.
Line 25: This paragraph is totally void of any reference while the previous international discussions on suburban village migrations are plenty. Please re-write your theoretical background and demonstrate you have read previous works.
Figure 1: Please add legends to all features in the figure.
Line 122: These methods should be linked to specific research questions (posed in introduction). Without research questions, readers will not be able to figure out what you are going to do and be lost in your description of technical details.
Line 176: These arguments need previous literature to support. For example, are there any studies using the same clustering methods? What are the studies using other alternatives? What are the advantages and dis-advantages and justify your choice.
Figure 5 and 6: Before showing these results, it is important to also have a figure to show the functioning zones of the study areas. Activity intensity level alone does not provide any insights for readers.
Line 300: Please move conclusions to the end of the paper and make clear your theoretical and methodological contributions in the conclusions (a case study is not counted as a significant contribution – it is on the broader picture how you contribute to the academia)
Discussions: Again, this part fails to engage any previous works on using big data/planning support methods in supporting urban planning (which are plenty). The impacts and values of your work will be compromised if you cannot convince the readers that your research has value-added to previous works, such as using big data and planning support methods to improve planning practices:
https://www.tandfonline.com/doi/abs/10.1080/10630732.2017.128501
Author Response
Dear Sir or Madam
It's my pleasure and honour to come across a reviewer like you. As you may guess, I am a new scholar. Your suggestion has played a great role in enriching my paper and my work. Your guidance has also helped me to learn how to complete an academic paper better. Thank you very much for your advice and help, and I agree with your comments on my article totally and sincerely. I have tried my best to revise my paper that follow your evaluation. I hope that I can meet the requirements of a complete and standard paper. The following is my reply to you:

Reviewer 2 Report
The article is quite interesting and the topic is worthy of investigation. The research methodology is quite innovative and in appearance rigorous. Below I detail issues that I find may be improved.
REVIEW OF THE STATE OF THE ART
The review of the state of the art is quite scarce (only 18 references) and homogeneous. Authors also go in the introduction too quickly to explain their case study. In my opinion the authors should include at least 5-10 international references more to introduce the article illustrating the importance of the topic, existing methodologies, gaps that need to be filled in the current field of research, etc.
In addition the authors state in the introduction “Traditional methods, including social investigation, questionnaires,interviews, tracking records, etc., rely on the sampling principles in statistics, which uses a finite random sample to estimate the overall characteristics. However, it is difficult to accurately obtain the activities and spatial characteristics of rural population in a complex environment, and the results are also difficult to guide urban and rural planning locally.” This statement is too ambitious since there exists many good spatial planning analysis methods for suburban areas in the scientific literature (see e.g. spatial suburban patterns of growth analysis in S. GarcĂa-AyllĂłn, Rapid development as a factor of imbalance in urban growth of cities in Latin America: A perspective based on territorial indicators. Habitat International 58, 2016, 127-142.)
CONCLUSIONS AND DISCUSSION
It is quite confusing to see discussion after conclusions. Their order should be inverted. In addition, the structure of discussion is not very scientific. Authors should explain here differences/coincidences/improvements of results in relation to previous studies in the area or field of research, limitations of the methodology developed, possible future lines of research, etc. By the other hand too few numerical results are included in the conclusions.
MINOR ISSUES
- Review carefully the manuscript format with the template of journal (references, formulas, figures, etc.)
- Review Table 3 format to fit it better in the sheet
Author Response
Dear Sir or Madam
It's my pleasure and honour to come across a reviewer like you. I am a new-comers for academic. Your suggestion has played a great role in enriching my paper and my work. Your guidance has also helped me to learn how to complete an academic paper better. Thank you very much for your advice and help, and I agree with your comments on my article totally and sincerely. I have tried my best to revise my paper that follow your evaluation. I hope that I can meet the requirements of a complete and standard paper. The following is my reply to you:

Reviewer 3 Report
The paper presents a study of spatial characterization of suburban/rural villages near a metropolitan area in China. The analysis is conducted mainly based on mobile phone data and previous surveys of the residents. In general, the idea of the paper has some interest and potential benefit for the region, as mentioned also in the study, however, in its current presentation, the paper comes with serious flaws. If those were revised, the paper could possibly be given higher value. Specific comments and suggestions for improvement are as follows:
There is almost no related works discussion in the paper which makes it even harder to evaluate the novelty and significance of the work. Although perhaps the fact that the study is specific to a particular area makes the paper novel enough, it has to be argued that the proposed methodology has some originality to offer for the scientific community
English has significant weaknesses throughout the presentation. In many cases, complete sentences lack verbs or natural flow. That makes the paper hard to read. A careful revision in terms of language is necessary.
In 2.3.1, it is not clear at all how handover information is used for increasing positioning accuracy. In general, the explanation of the use of mobile phone data is very weak. It is essential to explain exactly the type of data used from th mobile operator and to give more details on the methodology applied on it.
Many times in the paper, terms used are not explained before hand, making the ideas and methodology hard to follow. For example, in 2.3.1, the authors mention "settlement locations" and "grid hotspots" without first defining what a settlement is and how a grid hotspot is constructed. In another instance, in 2.3.2, there is no explanation about resident population data and rural field survey data. When multiple data types and datasets are used, it is essential to dedicate a section to clearly present those.
In 2.3.2, authors make an assumption about 10 days out of 30 for considering a user as a resident. Where is the assumption based?
What is i in equation (1)?
What is 672 in equation (4)?Explain better count of activities as well.
What is 48 in equation (5)?
The maps and graphs in the analysis results section are very interesting, along with the related discussion. However, with the weaknesses of the overall methodology presentation, it is very hard to assess those results.
For completeness, it is essential to explain the 9 villages categories, and the 5 service categories as well as what it means (qualitatively) to have a bimodal or trimodal or fluctuant activity curve.
Author Response

(The authors gave the same response as above.)

Round 2
Reviewer 1 Report
I appreciate the authors' efforts in improving the paper in this version. I feel that some language improvements can be used and I will leave this task to the editor.
Author Response
Dear Sir or Madam
Thank you for your advice. In this round of revision, I just added a small amount of content about positioning method of cellphone signalling which was a suggestion from other co-reviewer. With your help, I had learned a lot of methods and techniques for writing papers. I benefited greatly from this process.
The following is my reply to you:

Reviewer 3 Report
Thank you for taking into account our comments and for working so hard to improve and update the manuscript. It is obviously improved compared to its previous status, however, I stil feel that a lot of the information regarding the use and analysis of the mobile phone data is vaguely explained. There are many questions arising from your presentations, such as, what do you define a stop location, how do you decide which locations are stop locations? There is a o lot of research done to detect and define stop locations from mobile phone data, it is not a trivial issue and this paper treats it very weakly, although it is often implied that it is an important part of the work. I strongly recommend taking extra care on addressing that. English is better now, but still could be improved.
Author Response
Dear Sir or Madam
Thank you for your advice. In this round of revision, I have read a lot of relevant literature, and added some content about positioning method of cellphone signalling data. With your help, I had learned a lot of methods and techniques for writing papers. I benefited greatly from this process. The following is my reply to you:
